# The Gut Microbiome and Gastrointestinal Toxicities in Pelvic Radiation Therapy: A Clinical Review

**DOI:** 10.3390/cancers13102353

**Published:** 2021-05-13

**Authors:** Byeongsang Oh, Thomas Eade, Gillian Lamoury, Susan Carroll, Marita Morgia, Andrew Kneebone, George Hruby, Mark Stevens, Frances Boyle, Stephen Clarke, Brian Corless, Mark Molloy, David Rosenthal, Michael Back

**Affiliations:** 1Northern Sydney Cancer Centre, Royal North Shore Hospital, St Leonards, NSW 2065, Australia; Thomas.Eade@health.nsw.gov.au (T.E.); Gillian.Lamoury@health.nsw.gov.au (G.L.); Susan.Carroll@health.nsw.gov.au (S.C.); Marita.Morgia@health.nsw.gov.au (M.M.); Andrew.kneebone@health.nsw.gov.au (A.K.); george.hruby@health.nsw.gov.au (G.H.); mjmstevens@icloud.com (M.S.); stephen.clarke@sydney.edu.au (S.C.); bcorless@shoalhaven.net.au (B.C.); Michael.Back@health.nsw.gov.au (M.B.); 2The Mater Hospital, Sydney, NSW 2060, Australia; franb@bigpond.net.au; 3Faculty of Medicine and Health, University of Sydney, Sydney, NSW 2006, Australia; 4Bowel Cancer and Biomarker Laboratory, Faculty of Medicine and Health, University of Sydney, Sydney, NSW 2065, Australia; m.molloy@sydney.edu.au; 5Harvard Medical School, Boston, MA 02215, USA; drose@huhs.harvard.edu

**Keywords:** gut microbiome, radiotherapy, chemoradiotherapy, cancer, gastrointestinal toxicities

## Abstract

**Simple Summary:**

A substantial proportion of cancer patients receive radiotherapy (RT) during their cancer trajectory. One of the most challenging pelvic RT-related toxicities are gastrointestinal (GI) toxicities (e.g., abdominal pain, rectal bleeding, faecal incontinence, and diarrhoea) which impair the quality of life (QoL) of patients. Mounting evidence suggests that gut microbiota plays a pivotal role in health and disease, including cancer. Our current clinical review aims to assess the impact of RT on gut microbiota and GI toxicities in cancer patients to provide useful information, in addition to standard care, for clinicians and patients.

**Abstract:**

Background: Gastrointestinal (GI) toxicities are common adverse effects of pelvic radiotherapy (RT). Several recent studies revealed that toxicity of RT is associated with dysbiosis of the gut microbiome. Method: A literature search was conducted in electronic databases Medline, PubMed, and ScienceDirect, with search terms “microbiome and/or microbiota” and “radiotherapy (RT) and/or chemoradiation therapy (CRT)” and “cancer”, and the relevant literature were selected for use in this article. Results: Eight prospective cohort studies were selected for review with a total of 311 participants with a range of 15–134 participants within these studies. The selected studies were conducted in patients with gynaecological (*n* = 3), rectal (*n* = 2), or prostate cancers (*n* = 1), or patients with various types of malignancies (*n* = 2). Three studies reported that cancer patients had significantly lower alpha diversity compared with healthy controls. Seven studies found that lower alpha diversity and modulated gut microbiome were associated with GI toxicities during and after pelvic RT (*n* = 5) and CRT (*n* = 2), whereas one study found that beta diversity was related to a complete response following CRT. Two further studies reported that fatigue was associated with dysbiosis of the gut microbiome and low alpha diversity during and after RT, and with dysbiosis of the gut microbiome and diarrhoea, respectively. Conclusion: Gut microbiome profiles are associated with GI toxicities and have the potential to predict RT/CRT-induced toxicities and quality of life (QoL) in patients undergoing those treatments. Further robust randomized controlled trials (RCTs) are required to elucidate the effect of gut microbiome profiles on RT-related adverse effects and responses to RT.

## 1. Introduction

Radiotherapy (RT) is a standard cancer treatment for both curative and palliative patients and is often combined with chemotherapy to treat cancer, prevent recurrence of cancer, and improve quality of life (QoL) [1]. Approximately 50 percent of all cancer patients receive RT during their cancer trajectory [2]. Despite the benefits of RT in oncology being well established, RT-induced toxicities may detract from the therapeutic ratio. One of the most challenging pelvic RT-related toxicities are gastrointestinal (GI) toxicities (e.g., abdominal pain, tenesmus, rectal bleeding, faecal incontinence, and diarrhoea) which impairs the QoL of patients receiving RT in the abdominopelvic regions. Severe GI toxicities not only affect cancer patient’s QoL but may add to the cost of medical treatment, including additional use of analgesics and pain medication, prolonged hospital stays, and the interruption of cancer treatment [3,4]. However, as yet there are no effective strategies to predict and/or proactively manage RT treatment-induced GI toxicities. 

Growing evidence suggests that the gut microbiome plays a critical role in health and disease, and dysbiosis of gut microbiota can be a contributing factor in carcinogenesis [5,6,7]. For instance, *Helicobacter pylori*, a gut bacteria, is well recognized for increasing the risk of developing gastric ulcers and cancer [8]. Recently, several studies reported that RT induces alterations of gut microbiota profiles in cancer patients, and that dysbiosis of gut microbiota was related to the severity of RT-related toxicities, including mucositis, diarrhoea, pain, and fatigue [9]. 

The major changes in gut microbiota were reduced diversity of *Firmicutes* and *Bacteroidetes*, and increased *Proteobacteria* [10]. These findings were consistent with preclinical studies which revealed that dysbiosis of gut bacteria is related to these toxicities. Diversity, comprising richness and evenness of distribution of taxa within a given faecal sample, is a measurable outcome commonly used in microbial studies [11]. Recent studies suggested that high levels of alpha diversity in gut bacteria are related to low adverse events during RT treatment and higher response rates for cancer patients receiving immunotherapy [12].

Other studies have proposed that the gut microbiota profile can be applied as a non-invasive biopsy to predict radiotherapy-associated toxicity [13]. Besides treatment-related toxicities, several recent studies demonstrated that gut microbiota is associated with favourable responses to cancer treatments including chemotherapy [14], radiotherapy [10], and immunotherapy [13,15,16]. As a result, faecal microbiota transplantation (FMT) was developed and is currently under investigation in several countries [9,15,17,18,19]. 

To date, no literature reviews have examined the impact of RT on gut microbiota in relation to GI toxicities in cancer patients. Most previous reviews included both preclinical and clinical studies, and attempted to elucidate the underlying mechanisms of dysbiosis of gut microbiota in cancer pathogenesis [6,20,21,22], but did not focus on cancer treatment-related toxicities during RT. Hence, our current brief review of clinical studies assesses the impact of RT on gut microbiota and GI toxicities in cancer patients to provide useful additional information for clinicians and patients.

**Method:** A literature search was conducted using electronic databases Medline, PubMed, and ScienceDirect, with the main search terms “microbiome and/or microbiota” and “radiotherapy (RT) and/or chemoradiotherapy (CRT)” and “cancer”. Inclusion criteria in the searches were: clinical trials conducted with adults (> 18 years) and published in English. References of the included studies were carefully reviewed for relevant papers that would have been missed by electronic searches. The search strategy was performed for studies published up to September 2020.

**Results:** A total of 987 studies were initially identified from the three electronic databases (Medline, PubMed, and ScienceDirect). After an in-depth evaluation and screening of titles and abstracts, 73 articles remained to be further assessed for eligibility for inclusion in the study. Eight studies were included in the review (Figure 1, and Table 1 and Table 2).

### 1.1. Characteristics of Clinical Studies

The eight prospective studies included total of 311 participants with a range of 15–134 within each study, with 182 in the RT (intervention) group, 109 in the CRT (intervention) group, and 20 in the healthy control group. Studies were conducted on participants with gynaecological (*n* = 3) [23,24,25], rectal (*n* = 2) [26,27], prostate (*n* = 1) [28], and various types of cancer (*n* = 2) [29,30]. Two studies each were conducted in China [23,30], Korea [24,26], and the USA [25,27], and one study each in France [29] and the UK [28]. The earliest RCT was published in 2008 [29]. Four studies were two-arm prospective observational studies [24,28,29,30], while the other four studies were one-arm [23,25,26,27,28]. Total RT treatments comprised 25 sessions (50 Gy, 1.8–2.0 Gy/day, 5 times/week, 5 days) in five studies, while one study each varied in RT treatment dosage with 70–75 Gy for prostate cancer, 50–60 Gy for pelvic lymph nodes [28], and 51–53 Gy for rectal cancer [27]. One study did not describe RT dosage level [25]. Studies were conducted with patients receiving RT (*n* = 5) and CRT (*n* = 3) [25,26,27]. The primary outcome measures for GI toxicities included enteritis [23], enteropathy [24,28], bowel function [25], diarrhoea [29], diarrhoea-related fatigue [30], CRT-related fatigue [27], and response to RT [26]. Seven studies analysed the gut microbiome profile with the 16S ribosomal RNA (16S-rRNA) gene sequencing method (*n* = 7) and one study with denaturing gradient gel electrophoresis (DGGE). Interestingly, analysis of gene sequencing regions of 16S rRNA varied across studies: V1–V2 [24,26,28] (*n* = 3), V4 [23,25] (*n* = 2), V3 [30] (*n* = 1), and V3–V4 [27] regions of 16S rRNA (*n* = 1), and 16S rRNA position 968–1401 in *E. coli* [29] (*n* = 1).

### 1.2. Alpha Diversity of the Gut Microbiome before Treatment: Cancer Patients vs. Healthy Control

Of eight individual studies (RT (*n* = 5) and CRT (*n* = 3)) [23,24,25,26,27,28,29,30], three studies [24,29,30] compared alpha diversity of the gut microbiome between cancer patients and healthy control groups before undergoing RT. Three studies consistently reported that, when compared with controls, cancer patients had significantly lower alpha diversity.

### 1.3. Impact of RT/ CRT on the Gut Microbiome

Of eight individual studies [23,24,25,26,27,28,29,30], seven studies compared the impact of pelvic RT (*n* = 5) [23,24,28,29,30] and CRT (*n* = 2) [25,27] on alpha diversity and found that diversity decreased after pelvic RT and CRT, whereas one study did not report any findings [26].

### 1.4. GI Toxicities Induced by RT and the Gut Microbiome

Of five individual studies [23,24,28,29,30], four studies compared alpha diversity in patients either with or without diarrhoea and found that there was significantly lower alpha diversity in patients with diarrhoea [23,24,29,30]. Two studies assessed the relationship between alpha diversity and severity of enteritis and found that lower diversity was consistently associated with RT-related enteritis [28,30]. One study assessed RT-related symptoms of fatigue and diarrhoea and suggested that diarrhoea significantly increased fatigue scores at the third and fifth week of RT (*p* < 0.01), whereas fatigue scores of patients with no diarrhoea increased slightly [30].

A recent study conducted on patients with prostate cancer in the UK showed that patients with radiation enteropathy (RE) have higher counts of *Roseburia, Clostridium IV*, and *Faecalibacterium (p* < 0.05), and reported that intestinal mucosa cytokines (IL7, IL12/IL23p40, IL15, and IL16) were related to radiation enteropathy and inversely correlated with counts of *Roseburia* and *Propionibacterium* [28]. This study also reported that decreased SCFA production was significantly associated with enteritis [28]. Two studies conducted in China consistently demonstrated that RT-induced diarrhoea was related to a higher ratio of *Firmicutes* to *Bacteroides* [23,30]. A study of patients with cervical cancer (*n* = 18) found that genera *Serratia, Bacteroides*, and *Prevotella_9* were the most abundant in RE patients, while the proportion of *Bacteroides* was markedly reduced (21.23% vs. 43.83%, *ρ* = 0.004) [23]. Other minor genera that were significantly less abundant in RE patients were *Blautia* (*ρ* = 0.010) and *Ruminococcaceae_UCG-003* (*ρ* = 0.048) [23]. Another study in patients with both cervical and colorectal cancer (*n* = 11) compared patient groups (no diarrhoea vs. diarrhoea) following RT and found significantly higher levels of *Alistipes*, *Bacteroides*, *Clostridium_XI*, *Erysipelotrichaceae*, *Escherichia*, *Lachnospiracea*, and *Megamonas* in patients who developed diarrhoea, whereas *Clostridium_XIVa* and *Sutterella* were significantly lower in these patients [30]. Two studies found that, at the phylum level, *Actinobacteria* was related to cancer and RE [24,29]. Another study of patients with gynaecological cancer (*n* = 9) in Korea reported that *Actinobacteria* was 30 times higher in cancer patients than in healthy individuals [24]. An earlier study in France conducted with patients with abdominal cancer (*n* = 10) found that patients with diarrhoea had increased levels of *Actinobacteria* after RT, although its presence was not detected in patients without diarrhoea prior to RT [29].

### 1.5. GI Toxicities Induced by CRT and the Gut Microbiome

Recently, three studies examined the effect of CRT on the gut microbiome [25,26,27]. One study assessed the effect of CRT on fatigue and the gut microbiome in patients with rectal cancer (*n* = 29) in the USA and reported that decreased alpha diversity of the gut microbiome increased fatigue after CRT [27]. They found that the relative abundance of *Bacteroidetes* was significantly higher in patients with fatigue compared with those who were not fatigued (*p* < 0.05). They also found that *Proteobacteria*, *Firmicutes*, and *Bacteroidetes* were the dominant phyla and *Escherichia*, *Bacteroides*, *Faecalibacterium*, and *Oscillospira* were most abundant at the genus level (*p* < 0.05) in patients with fatigue, while the *Firmicutes* phylum, including members of the *Lactobacillaceae* family such as *Lactobacillus* genus, were significantly enriched in patients who were not fatigued (*p* < 0.05) [27]. Another study in the USA assessed the effect of CRT on bowel function and the gut microbiome in patients with advanced cervical cancer (*n* = 29) and found that alpha diversity and the relative abundance of *Clostridiales* had declined over time, while other phyla were relatively stable [25]. This study reported that higher alpha diversity was correlated with lower GI toxicities during and after CRT, but not with baseline diversity. Another recent study [24], conducted with preoperative patients with rectal cancer (*n* = 45), assessed the impact of CRT on the gut microbiome and reported that seven patients (16%) demonstrated pathologically complete responses (CR), and 38 patients (84%) showed non-CR after preoperative concurrent CRT. They also found a significant difference in beta diversity (*p* = 0.028) between patients with CR and non-CR, but not in alpha diversity. In this study, *Cyanobacteria*, the family of *Corynebacteriaceae* and *Clostridiaceae*, were dominant in the CR group, while *Bacteroidales (Bacteroidaceae, Rikenellaceae, Bacteroides)* were relatively more abundant in patients with non-CR.

## 2. Discussion

A significant finding in this review is that low alpha diversity and dysbiosis of the gut microbiome are associated with GI toxicities induced by pelvic RT. Although a number of studies (*n* = 8) were conducted with heterogeneous cancer populations, viz., rectal cancer (*n* = 2), cervical cancer (*n* = 2), prostate cancer (*n* = 1), and mixed cancer groups (*n* = 3), seven studies examined the relationship between bacterial diversity and composition of the gut microbiome. In these studies, GI-related adverse events were consistently found to be related to low alpha diversity and dysbiosis of the gut microbiome following RT [23,24,25,27,28,29,30]. Of those studies, three compared the diversity of the gut microbiome in cancer patients and healthy controls, and identified that, compared with healthy controls, alpha diversity was lower in cancer patients [24,29,30]. These findings are similar to previous reviews conducted with various medical conditions including cancer [31,32], irritable bowel syndrome [33], Crohn’s disease [34,35], diabetes [36,37], obesity [38], and chronic pain [39]. Several studies examined the relationship between alpha diversity in the gut microbiome during immunotherapy in advanced melanoma [40,41], lung [16], and liver [42,43] cancers, and found that alpha diversity was associated with a positive response to immunotherapy, as measured by progression-free survival (PFS) and overall survival (OS). Notably, a recent breakthrough reported that alpha diversity of gut bacteria at baseline correlated with an improved recurrence of free survival (RFS) and OS, resulting in the proposal that alpha diversity has a potential use as an independent predictor of survival in cervical cancer patients receiving CRT [12]. Nonetheless, one study addressed the issue of current outcome prediction using measurements of diversity, which in most studies are measured based on species-level or operational taxonomic units (OUT)–level diversity [44]. Due to the lack of knowledge of the functional differences in multiple taxonomic levels of diversity, there is a need to assess and examine these in future studies.

Despite advances in radiation technology, enabling it to deliver a radiation beam precisely, and improvements in pharmacological interventions, to lessen RT-induced toxicities, pelvic RT-induced GI toxicities remain challenging [45,46]. Currently, there are limited strategies available to predict and to minimise GI toxicities induced by pelvic RT. For example, reducing RT dosage can lower GI toxicities, however, reduction of the dosage of RT can compromise its efficacy and may increase the risk of cancer recurrence. Considering the limitations in the management of GI toxicities during RT/CRT treatment, our findings provide a novel proposal that alpha diversity in the gut microbiome has the potential to be used as a predictive biomarker of GI toxicities in pelvic RT, to minimize RT-induced toxicities, and to improve QoL of patients.

Furthermore, cancer-related fatigue (CRF) is one of the most prevalent symptoms of cancer. It has been reported that up to 90% of cancer patients receiving RT experience CRF [47]. Despite a recent study reporting that fatigue is related to diarrhoea [48], fatigue is a complex symptom that relates to multiple factors including anaemia, insomnia, pain, dyspnoea, loss of appetite, depression, and anxiety [47,49]. Two studies [27,30] included in our review reported that the gut microbiome is related to fatigue. One of these reported that diversity in the microbiome was lower in patients with diarrhoea, and that fatigue scores increased in patient groups with diarrhoea [30], but the other study did not examine the relationship between diarrhoea and fatigue [27]. Given that a limited number of studies have investigated the relationship between fatigue and diarrhoea related to the gut microbiome, we believe that further exploration of this relationship is warranted in future studies.

Our review provides guidance for future studies. Interestingly, the effect of RT on the gut microbiome in patients with non-pelvic RT is yet to be explored and a proposed study, for example, would be to assess the toxicities of RT on the gut microbiome in patients with breast cancer. Previous studies have examined the microbiome in breast cancer patients who had not undergone RT and identified that there were differences in the microbiome between cancer cells and non-cancer cells in breast tissue [50,51]. With regard to these findings, it would be worthwhile, in the future, to examine potential relationships between the gut microbiome and the microbiome in breast tissue.

Moreover, it will be worthwhile to examine the relationship between the diversity of the gut microbiome and cancer biomarkers, as well as RT-induced toxicities. Future innovative studies are required to investigate the effects of the gut microbiome on responses to RT/CRT and survival rates and, to date, few studies in RT/CRT have examined these relationships. Despite several recent studies reporting a positive relationship between the diversity of the gut microbiome and clinical response to immunotherapies in patients with melanoma [13,15,40], these associations remain largely unexplored.

A recent study exploring the relationship between the gut microbiome and the CRT response found that beta diversity of the gut microbiome prior to CRT, but not alpha diversity, was related to a complete response to CRT [26]. Although seven studies consistently reported a relationship between lower alpha diversity and RT-related adverse events, a study examining the gut microbiome in rectal cancer patients that attempted to find a predictive biomarker for complete response after current chemoradiotherapy (CRT) reported that there was no significant difference in alpha diversity between complete response (CR) (*n* = 7) and non-CR (*n* = 38) groups, whereas beta diversity was significantly higher in the CR group [26]. Similarly, in a study conducted in advanced lung cancer patients during chemotherapy (CTX) [52], it was reported that the diversity of the gut microbiome was not related to the efficacy of CTX. In contrast, several studies examined the relationship between alpha diversity in the gut microbiome during immunotherapy in advanced melanoma [40,41], lung [16], and liver [42,43] cancers, and found that alpha diversity was associated with a positive response to immunotherapy, as measured by progression-free survival (PFS) and overall survival (OS). Hence, we believe that there is a need for further studies to examine the relationship between alpha diversity and RT-related adverse effects in patients receiving CRT.

Although our review highlights findings that the diversity of the gut microbiome is related to GI toxicities in RT/CRT, the direction of this relationship is unclear. A question remains as to whether GI toxicity induced by RT leads to further alterations in diversity and composition of the microbiome, or whether dysbiosis of the microbiome leads to GI toxicity. Well-designed RCTs are needed to examine effective interventions that can improve and/or reduce impairment of microbiome diversity and composition, and confirm the role of the gut microbiome in RT-related toxicities and response.

The current review highlights several limitations of the studies included here. Of the eight studies reviewed, four were conducted with small sample sizes, ranging from 9 to 18 participants [23,24,29,30], and three ranging from 29 to 45 participants with no appropriate sample size calculation [25,26,27]. Although one study of prostate cancer patients had a larger sample size (*n* = 134) [28], it did not report effect size, thus limiting its generalizability. The calculation of effect size in microbiome clinical studies was recognized as a challenging issue because of the complexity of the microbiome analysis. In the past, researchers have been unable to agree on a common acceptable method of determining the appropriate effect size in clinical studies [53], however, recently, with the accumulation of 16S rRNA gene sequencing data, estimation models for power and sample size have been proposed for use in clinical trial design [54,55]. Given that the current review has demonstrated the association between the gut microbiome and GI toxicities, future robust, well-designed RCT microbiome studies, with adequate effect size, are needed to confirm the nature and direction of this relationship, as well as clinical responses to RT and patient survival. Another limitation in the reviewed studies related to poor statistical control for potentially confounding factors in the data analyses, despite providing descriptions of inclusion and exclusion criteria in an attempt to minimize some of the confounding effects. In the design of future microbiome studies, researchers should consider collecting additional data such as participant’s body mass index (BMI), lifestyle (exercise, diet, smoking status), mental and emotional status (anxiety, depression, stress), and the co-occurrence of other chronic medical conditions, in order to control for confounding factors in the data analysis. For example, several studies recently reported that the composition of the gut microbiome and the ratio of *Firmicutes* to *Bacteroidetes* were related to obesity [33,56,57]. However, the studies in this review did not control for BMI during data analysis, which raises questions about the reliability of those results as it is possible that the relationship between lower diversity of the microbiome and GI toxicities resulted from participants’ obesity, rather than from RT treatment.

Other recent studies have reported that lifestyle factors, including exercise [58], emotional status [59] (anxiety, depression, and stress), and chronic disease (pain, Crohn’s disease, diabetes, insomnia, and hypertension) are associated with dysbiosis of the gut microbiome [34,35,36,60,61,62], and these factors were not controlled for in the reviewed studies. Another limitation in all of the reviewed studies was the use of, mainly, the 16S rRNA gene sequencing method and in one study, denaturing gradient gel electrophoresis (DGGE). The 16S rRNA and shotgun metagenomic sequencing methods are commonly used in other studies [15,40]. The 16S rRNA gene sequencing method, a form of amplicon sequencing, targets and reads a region of the 16S rRNA gene, which is found in all bacteria and archaea, and has the advantage of analysing taxonomic profiling, whereas shotgun metagenomic sequencing can read all genomic DNA in a sample and provide data on metabolic functional potential in addition to identifying taxonomy of the gut microbiome. However, 16S rRNA gene sequencing is commonly used in microbiome studies because of the well-established bioinformatics data analysis pipelines and its lower cost, approximately US$50–70 per sample, compared with shotgun metagenomic sequencing which costs around US$200–$300 per sample [63]. In order to generate more comprehensive microbial functional data, future studies would benefit from applying shotgun metagenomic sequencing methods.

Despite these limitations, the current literature review is, to our knowledge, the first to examine the impact of RT/CRT on GI toxicities and the gut microbiome profile in clinical observational studies, and provides meaningful information in order to reduce RT/CRT-induced GI toxicities for oncologists and cancer patients.

In conclusion, the current review demonstrates that dysbiosis of the gut microbiome is related to GI toxicities induced by RT/CRT and that gut microbiome profiles can be used as predictors of GI toxicities prior to RT, thereby potentially reducing the incidence of cancer recurrence. However, before these findings can be recommended as standard screening tools, further robustly designed studies with appropriate effect size are warranted.

## Figures and Tables

**Figure 1 cancers-13-02353-f001:**
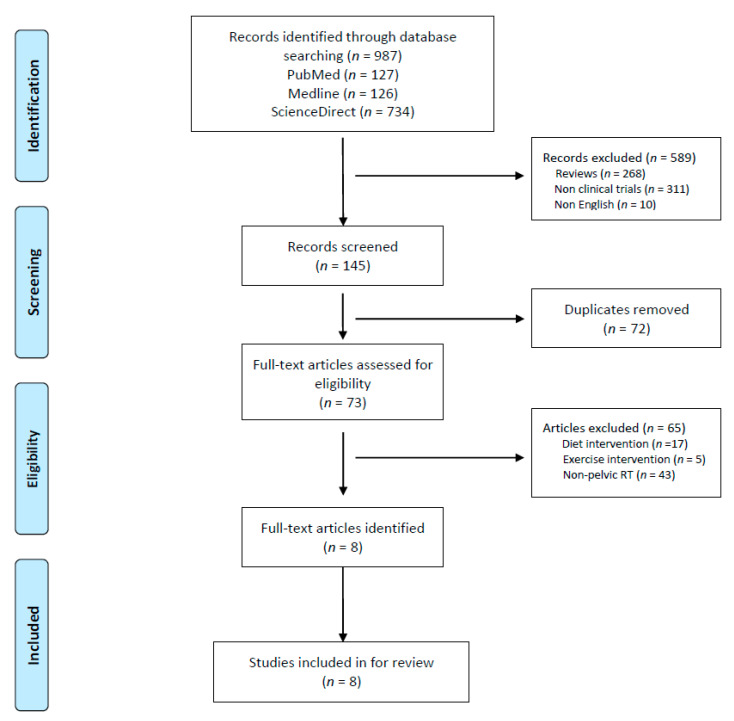
Preferred reporting items for systematic review and meta-analysis.

**Table 1 cancers-13-02353-t001:** Gut microbiome studies in RT/CRT.

Author Year Country	Study Subjects Sample Size Mean Age (Range)	Treatment Type	Outcome	Faecal Sample Collection	Microbiome Analysis Method	Conclusions
Gonzalez-Mercado et al. [27] 2020 USA	Rectal cancer patients (*n* = 29), mean age: 61 yrs (range 37–80 yrs)	CRT Total RT (51–53 Gy): 45 Gy in 25 fractions plus 6 to 8 Gy boost 5-FU (*n* = 17), oral capecitabine (*n* = 10)	Fatigue	3× (before, at the middle, and at the end)	V3/V4 region of the 16S rRNA	RT-associated perturbation of the gut microbiome composition may contribute to fatigue.
Jang et al. [26] 2020 Korea	Preoperative rectal cancer patients (*n* = 45)	CRT 50.0 Gy in 25 fractions (*n* = 4), 50.4 Gy in 28 fractions (*n* = 33), 54 Gy in 30 fractions (*n* = 8)	Response to RT: complete response (CR) (*n* = 7), non-CR (*n* = 38)	1× (prior to CCRT)	V1–V2 region of the 16S rRNA	Samples obtained before preoperative CCRT, differences in microbial community composition and functions were observed between patients with and without CR in rectal cancer.
Mitra, A. et al. [25] 2020 USA	Advanced cervical cancer (clinical stage IB1, IB2, IIA, IIB, IIIB, and IVA) (*n* = 35)	CRT (RT plus cisplatin)	Bowel function	4× (baseline and at weeks 1, 3, and 5)	V4 region of the 16S rDNA	Increased RT toxicity is associated with decreased gut microbiome diversity. Baseline diversity is not predictive of end-of-treatment bowel toxicity, but composition may identify patients at risk for developing high toxicity.
Ferreira et al. [28] 2019 UK	Prostate cancer (PCa) patients (*n* = 134), 1st cohort (*n* = 32), 2nd cohort (*n* = 87), 3rd colonoscopy cohort (*n* = 15), PCa (*n* = 9), healthy controls (*n* = 6) (range 63–79 yrs)	Conventionally fractionated RT: 70–74 Gy to prostate and seminal vesicles (35–37 fractions) or 64 Gy to prostate bed (32 fractions); 50–60 Gy to pelvic lymph nodes (35–37 fractions) Hypofractionated RT: 60 Gy to prostate and seminal vesicles or 55 Gy to prostate bed (20 fractions); 47 Gy to pelvic lymph nodes	Enteropathy	6× (at baseline and at 2/3 weeks, 4/5 weeks, 12 weeks, 6 months, and 12 months post-RT)	V1–V2 region of the 16S rRNA	An altered microbiota associates with early and late radiation enteropathy, with clinical implications for risk assessment, prevention, and treatment of RT-induced side-effects.
Wang et al. [23] 2019 China	Cervical cancer stage II–IV (*n* = 18) mean age: 57 yrs (range 30–67 yrs)	RT: 50.4 Gy in 1.8 Gy/fraction	Enteritis (*n* = 10), non-enteritis (*n* = 8)	2× (pre- and post-RT)	V4 region of the 16S rRNA	Gut microbiota can offer a set of biomarkers for prediction, disease activity evaluation, and treatment selection in RE.
Wang et al. [30] 2015 China	Patients with colorectal, anal, cervical cancer (*n* = 11), cervical cancer (*n* = 8), female anal cancer (*n* = 1), male colorectal cancer (*n* = 2) (range 41–64 yrs), healthy controls (*n* = 4)	RT: 1.8–2.0 Gy/day, 5 times/ week, 5 weeks	Fatigue measured with the MFI-20 and diarrhea measured with the CTCAE, diarrhea (*n* = 6), no diarrhea (*n* = 5)	2× (before and just after RT treatment)	V3 region of the 16S rRNA	In patients with diarrhea, fatigue scores significantly increased at both the third and fifth week of radiotherapy (*p* < 0.01), while those of patients with no diarrhea increased slightly. The microbial composition was also significantly different at the genus level prior to and post-radiotherapy in both groups of cancer patients.
Nam et al. [24] 2013 Korea	Gynecological cancer (*n* = 9) (age: 35–63 yrs), cervical cancer (*n* = 7), endometrial cancer (*n* = 2), healthy controls (*n* = 5)	RT: 50.4 Gy, 1.8–2.0 Gy/day, 5 times/week, 5 weeks	Diarrhea (*n* = 8), no diarrhea (*n* = 1)	4× (before, after the first radiotherapy, at the end, and follow- up after treatment)	V1/V2 region of the 16S rRNA	Overall gut microbial composition was gradually changed after treatment of pelvic RT. Dysbiosis of the gut microbiome was linked to health status.
Manichans et al. [29] 2008 France	Abdominal cancer (*n* = 10), cervical cancer (*n* = 1), endometrial cancer (*n* = 4), rectum cancer (*n* = 4), uterus cancer (*n* = 1), healthy controls (*n* = 5)	RT: 1.8–2.0 Gy/day, 5 times/week, 5 weeks	Diarrhea (*n* = 6), no diarrhea (*n* = 4)	4× (before, during, at the end, and 2 weeks after treatment)	16S rRNA (region 968–1401, positions in *E. coli* measured with DGGE)	Patients exhibiting diarrhea showed a progressive modification in their microbial diversity. Study indicates that diarrhea during RT may be linked to their initial microbial composition.

CRO: Clinician-reported outcomes, RTOG: The Radiation Therapy Oncology Group, CTCAE: Common Terminology Criteria for Adverse Events, UCLA-PCI: University of California, Los Angeles Prostate Cancer Index, MFI-20: multidimensional fatigue inventory, RE: radiotherapy enteritis, RT: radiotherapy, CRT: chemoradiotherapy, DGGE: denaturing gradient gel electrophoresis, 16S rRNA: 16 Svedberg unit (S) rRNA.

**Table 2 cancers-13-02353-t002:** Gut microbiome in response to RT/CRT.

Study	Intervention/GI Toxicites	Diversity			Phylum Level					Other Taxonomic Level (Order, Family, Genus and Species)
		α-Diversity	β-Diversity	Ratio Firmicutes/ Bacteroidetes	Bacteroidetes	Firmicutes	Proteo Bacteria	Fuso Bacteria	Actino Bacteria	
Gonzalez-Mercado et al. [27] 2020	CRT	**↓**								
	Fatigue			**↓**	**↑**	**↑**	**↑**			*Escherichia* (genus) ↑*,* *Bacteroides* (genus) ↑, *Faecalibacterium* (genus) ↑*,* *Oscillospira* (genus) ↑
	Non-fatigue									*Lactobacillaceae* (family) ↑, *Lactobacillus* (genus) ↑
Jang et al. [26] 2020	CRT complete response	NS	S							Cyanobacteria (phylum) ↑, Corynebacteriaceae (class) ↑, *Clostridiaceae* (family) ↑
	Non-complete response									Bacteroidales (order) ↑, *Bacteroidaceae* (family) ↑, *Rikenellaceae* (family) ↑, *Bacteroides* (genus) ↑
Mitra et al. [25] 2020	CRT	**↓**								Clostridiales (order) ↓
	Bowel function									
Ferreira et al. [28] 2019	RT	**↓**								
	Enteropathy									*Roseburia* (genus) ↑*,* *Clostridium IV* (genus) ↑, *Faecalibacterium* (genus) ↑
Wang et al. [23] 2019	RT									
	Enteritis	**↓**	**↑**		**↓**	**↓**	**↑**			Gammaproteobacteria (class) ↑*, Bacteroides. Coprococcus* ↓, Enterobacteriales (order) ↑*,* Oceanospirillales (order)↓, *Enterobacteriaceae* (family) ↑, *Phyllobacteriaceae* (family) ↑, *Beijerinckiaceae* (family) ↑, *Bacteroidaceae* (family) ↓, *Ruminococcaceae* (family) ↓, *Serratia* (genus) ↑, *Bacteroides* (genus)↑, *Prevotella_9* (genus) ↑
	Non-enteritis				**↑**		**↓**			*Enterobacteriaceae* (family)↑, *Phyllobacteriaceae* (family) ↑, *Eijerinckiaceae* (family) ↑, *Bacteroidaceae* (family) ↑, *Ruminococcaceae* (family) ↑
Wang et al. [30] 2015	Cancer vs. healthy	**↓**								*Faecalibacterium* (genus) ↑, *Clostridium_XI* (genus) ↑, *Roseburia* (genus) ↑, *Veillonella* (genus) ↑
	RT									*Bacteroides (genus)*↑, *Clostridium_XIVa* (genus) ↑, *Faecalibacterium* (genus) ↓, *Lachnospiracea* (family) ↓, *Oscillibacter* (genus) ↓, *Roseburia* (genus) ↓, *Streptococcus* (genus) ↓
	RT diarrhea	**↓**								*Alistipes* (genus) ↑, *Bacteroides* (genus) ↑, *Clostridium_XI* (genus) ↑, *Erysipelotrichaceae* (family) ↑, *Escherichia* (genus) ↑, *Lachnospiracea* (family) ↑, *Megamonas* (genus) ↑, *Clostridium_XIV* (genus) ↓, *Sutterella* (genus) ↓
Nam et al. [24] 2013	Cancer patients vs. healthy	**↓**			**↓**		**↓**	**↓**	**↑**	*Clostridiaceae* (family) ↑, *Ubacteriaceae* (family) ↑, *Prevotellaceae* (family) ↓, *Oscillospiraceae* (family) ↓, *Fusobacteriaceae* (family) ↓
	RT diarrhea	**↓**				**↓**		**↑**		*Eubacteriaceae* (family) ↓, *Fusobacteriaceae* (family) ↑, *Streptococcaceae* (family) ↑
Manichans et al. [29] 2008	Cancer patients vs. healthy									Significant microbial profile changes in patients with diarrhea during and after RT.
	RT Diarrhea	**↓**				**↑**			**↑**	

**↑****:** increased, **↓****:** decreased.

## Data Availability

Not applicable.

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
