# Peer review of "The Gut Microbiome and Gastrointestinal Toxicities in Pelvic Radiation Therapy: A Clinical Review"

_cancers, 2021, doi:10.3390/cancers13102353_

Round 1
Reviewer 1 Report
Oh et all provide a review of 8 studies on gut microbiome and interaction with RT. Topic selection is timely, and such analysis is clearly needed in evolving roles of RT and other therapies in cancer.
- Among drawbacks are absences of bias analysis from published studies, certainty of evidence could be better explored.
- Attempt at estimating effect size of reported studies could be done, although admittedly would not be easy, or even perhaps impossible: how much was QoL improved among those with prebiotics?
- Was there any differentiation of prebiotics, probiotics and synbiotics?
- Any interaction of effect of RT vs chemoRT?
- Please provide PRISMA flow diagram, and consider using PRISMA statement
Author Response
Reply to reviewer 1’s comments:
1. Among drawbacks are absences of bias analysis from published studies, certainty of evidence could be better explored.
Thank you for your comments. As the reviewer mentioned, our manuscript did not evaluate the risk of bias (ROB) in the reviewed studies. Quality appraisal of non-randomized studies (NRS) is complicated by the issue of heterogeneity in observational study designs (e.g., cohort, case-control, retrospective studies). Further, there is no robust method that is accepted as “gold standard” when evaluating ROB for NRS, despite methodological tools for assessing ROB in RCTs being well-established, e.g., the Cochrane Collaboration’s ROB Tool.
Although, we did not refer to biases arising from heterogeneity and other confounding factors in the methods and results sections, these issues were addressed in the discussion section of our manuscript.
2. Attempt at estimating effect size of reported studies could be done, although admittedly would not be easy, or even perhaps impossible: how much was QoL improved among those with prebiotics?
Thank you for understanding the complex issues related to estimation of effect sizes in gut microbiome studies. We agree with reviewer’s comments on the estimation of effect size and in our review referred to a current debate among biostatisticians relating to a lack of agreement on an appropriate model for calculating effect sizes in complex microbiome studies. We hope that the data presented in this manuscript will help researchers examine proposed models for estimation of power and sample sizes in the design of future studies.
3. Was there any differentiation of prebiotics, probiotics and synbiotics?
In terms of diet supplement, there are different categories of classification for prebiotics, probiotics and synbiotics. However, no studies to date have compared the beneficial effects of interventions with different dietary supplements (viz. prebiotics vs probiotics vs synbiotics) on gut bacteria during RT. Several studies have suggested that it may worthwhile comparing the effects of these interventions (prebiotics vs probiotics vs synbiotics) in future studies.
4. Any interaction of effect of RT vs chemo RT?
Thank you for this interesting question. We hypothesise that damage to gut bacteria will be greater in a concurrent chemotherapy plus RT group compared to an RT-only group. However, no studies to date have reported a comparison of the effects of these different interventions (chemo plus RT vs RT-only) on gut bacteria. It will be interesting to examine these effects in future investigations.
5. Please provide PRISMA flow diagram and consider using PRISMA statement.
We would like to thank the reviewer for this suggestion. Please find a PRISMA flow chart and statement in the manuscript as the reviewer suggested.
A total of 987 studies were initially identified from the three electronic databases (Medline, Pubmed and ScienceDirect). After an in-depth evaluation of screening titles and abstracts, 73 articles remained to be further assessed for eligibility to be included in the study. Eight studies were included in the review (Figure 1, and Table 1-2)
Reviewer 2 Report
The authors gave a novel insight into the impact on gut microbiome in cancer patients upon radiotherapy. The did extensive literature survey to find the relevance of gut microbiome and cancer. I have few minor comments to address.
- Since the authors propose Alpha diversity as a prediction marker for toxic effect upon RT in gut microbiome, but there is no much relevant evidences shown or any trained model to validate their proposal.
- In the discussion/introduction section, they need to add few more context about the importance of looking into alpha diversity and how it reflects the gut microbial consortia.
- A figure model where it expresses the impact of RT on gut microbiome in cancer patients can be added.
Author Response
Reply to reviewer 2’s comments
1. Since the authors propose Alpha diversity as a prediction marker for toxic effect upon RT in gut microbiome, but there is no much relevant evidences shown or any trained model to validate their proposal.
We agree with reviewer’s comments. It is a worthwhile suggestion to develop models for validation in future studies. Most of the original studies included in our review examined relationships in alpha diversity of gut bacteria with faecal samples collected before RT and RT-related GI toxicities. This data generated hypotheses about the nature of the relationships and provided insight into the role of gut bacteria in RT. In order to validate a model, researchers will need to conduct studies with the interventions (e.g. modulation of alpha diversity and composition of gut bacteria) during RT, after identifying low diversity of gut bacteria in cancer patients.
We therefore concluded that further robust randomized controlled trials (RCTs) are required to elucidate the effects of gut microbiome profiles on RT-related adverse effects and responses to RT.
2. In the discussion/introduction section, they need to add few more context about the importance of looking into alpha diversity and how it reflects the gut microbial
Thank you for the comments on alpha diversity. We have now amended the introduction and discussion sections of the manuscript to better reflect the significance of alpha diversity.
Please find this new sentence in the introduction section.
“Diversity, comprising richness and evenness of distribution of taxa within a given faecal sample, is a measurable outcome commonly used in microbial studies [1].
Recent studies have suggested that high levels of alpha diversity in gut bacteria is related to low adverse events during RT treatment and higher response rates for cancer patients receiving immunotherapy [2].”
Please find additional sentences in the discussion section.
Several studies examined the relationship between alpha diversity in the gut microbiome during immunotherapy in advanced melanoma [3, 4], lung [5] and liver [6, 7] cancers, and found that alpha diversity was associated with a positive response to immunotherapy, as measured by progression-free survival (PFS) and overall survival (OS). Notably, a recent breakthrough reported that alpha diversity of gut bacteria at baseline correlated with an improved recurrence of free survival (RFS) and OS, resulting in the proposal that alpha diversity has a potential use as an independent predictor of survival in cervical cancer patients receiving CRT [2]. Nonetheless, one study addressed the issue related to current outcome measurement of diversity, whereas most studies measured diversity based on species-level or operational taxonomic units (OUT) - level diversity [8]. While there is lack of knowledge on functional differences in multiple taxonomic levels of diversity, there is a need to assess and examine multiple taxonomic levels of diversity in studies.
3. A figure model where it expresses the impact of RT on gut microbiome in cancer patients can be added.
We would like to add a figure model to express the impact of RT on the gut microbiome in cancer patients as the suggested. However, it is a very challenging task to develop a model based on limited clinical trials, conducted with observational studies and non-homogenous cancer patient groups. Although, previous reviews were conducted on both clinical and pre-clinical studies, figure models presented were based only on preclinical studies. However, we will consider developing and presenting a figure model when the findings of our ongoing clinical trials examining the impact of RT on breast (n=50) and prostate (n=50) cancer patients in our cancer centre, are reported in full.
Reviewer 3 Report
The article titled "The Gut Microbiome and Gastrointestinal Toxicities in Pelvic Radiation Therapy: A Clinical Review" by Oh et al is a very important study as to aware patients who receive radiotherapy (RT) as part of their treatment schedule. The author covered most of the important published manuscripts and claimed "To date, no literature reviews have examined the impact of RT on gut microbiota in relation to GI toxicities in cancer patients", but there are a few more to cite and explain in the discussion.
- PMID: 28590950, PMID: 33436010
- Graphical representation of the tabular data is easier to understand. The author should consider plotting the data with appropriate statistical analysis.
With this modification, the review would be very impactful to the Journal as well as the researchers in this field.
Author Response
Reply to reviewer 3’s comments
- The author covered most of the important published manuscripts and claimed "To date, no literature reviews have examined the impact of RT on gut microbiota in relation to GI toxicities in cancer patients", but there are a few more to cite and explain in the discussion.
We would like to thank the reviewer for these comments and suggestions. As we described in the title, this paper is a brief review of clinical studies which aims to provide current evidence to inform clinicians and patients on specific adverse events in pelvic radiation therapy. Several review papers were published recently which have also included both clinical and preclinical studies. In our manuscript, we updated previous reviews to include additional studies published in 2017 and 2020, and in the introduction section described the significance of our review compared to previous clinical reviews. Please see the full paragraph.
“To date, no literature reviews have examined the impact of RT on gut microbiota in relation to GI toxicities in cancer patients. Most previous reviews included both preclinical and clinical studies, and attempted to elucidate the underlying mechanisms of dysbiosis of gut microbiota in cancer pathogenesis [9-11], but did not focus on cancer treatment-related toxicities during RT. Hence, our current brief review of clinical studies, assesses the impact of RT on gut microbiota and GI toxicities in cancer patients to provide useful additional information for clinicians and patients.”
Liu et al’s study included both clinical and preclinical studies [12]. Muls et al’s study was focused on women with gynaecology cancer [13], whereas our study included all cancer patient groups receiving pelvic radiotherapy.
However, as the reviewer suggested, we cited the additional two references in our manuscript.
- Muls A, Andreyev J, Lalondrelle S, et al Systematic Review: The Impact of Cancer Treatment on the Gut and Vaginal Microbiome in Women With a Gynecological MalignancyInternational Journal of Gynecologic Cancer 2017;27:1550-1559.
- Liu J, Liu C, Yue J. Radiotherapy and the gut microbiome: facts and fiction. Radiat Oncol. 2021 Jan 13;16(1):9. doi: 10.1186/s13014-020-01735-9. PMID: 33436010; PMCID: PMC7805150.
- Graphical representation of the tabular data is easier to understand. The author should consider plotting the data with appropriate statistical analysis.
We would also like to thank reviewer 3’s very important comments on our article and for the helpful suggestions to improve the quality of our paper. We would have preferred to add a graphical representation of the tabular data to make it easy to understand, as the reviewer suggested. However, it is a very challenging task to develop a graphical representation of very complex data presented as an operational taxonomic units (OUT) diagram with little quantitative data available, other than the presented P values. Further, as we stated in our discussion section, most clinical studies included in this review used the 16S rRNA gene sequencing method to measure the gut microbiome and assess the relative ratio, composition and diversity of gut bacteria from phylum to genus but not species level. This method cannot measure the total number of each gut bacteria, function and metabolites.
However, we will consider developing and presenting a figure model when the findings of our ongoing clinical trials examining the impact of RT on breast (n=50) and prostate (n=50) cancer patients in our cancer centre, are reported in full. We plan to analyse data using a shotgun metagenomic method instead of a 16sr RNA method.
I would like to thank again the editor and reviewers for providing us with constructive comments.
Round 2
Reviewer 1 Report
The authors sufficiently responded to my questions & provided PRISMA diagram.